# The Relevance of Implementing the Systematic Screening of Perioperative Myocardial Injury in Noncardiac Surgery Patients

**DOI:** 10.3390/jcm12165371

**Published:** 2023-08-18

**Authors:** Ekaterine Popova, Pilar Paniagua-Iglesias, Jesús Álvarez-García, Miquel Vives-Borrás, Aránzazu González-Osuna, Álvaro García-Osuna, Mercedes Rivas-Lasarte, Gisela Hermenegildo-Chavez, Ruben Diaz-Jover, Gonzalo Azparren-Cabezon, Montserrat Barceló-Trias, Abdel-Hakim Moustafa, Raul Aguilar-Lopez, Jordi Ordonez-Llanos, Pablo Alonso-Coello

**Affiliations:** 1Institut d’Investigació Biomèdica Sant Pau (IIB SANT PAU), 08041 Barcelona, Spain; raul.aguilar@vhir.org (R.A.-L.); palonso@santpau.cat (P.A.-C.); 2Centro Cochrane Iberoamericano, 08025 Barcelona, Spain; 3Department of Anesthesia and Pain Management, Hospital de la Santa Creu i Sant Pau, 08025 Barcelona, Spain; ppaniagua@santpau.cat (P.P.-I.); ghermenegildo@santpau.cat (G.H.-C.); rdiazj@santpau.cat (R.D.-J.); gazparren@santpau.cat (G.A.-C.); 4Department of Cardiology, Hospital Universitario Ramon y Cajal, 28034 Madrid, Spain; 5Department of Cardiology, Hospital de la Santa Creu i Sant Pau, 08025 Barcelona, Spain; mikivives@msn.com (M.V.-B.); rivaslasarte@gmail.com (M.R.-L.); abedmoustafa@gmail.com (A.-H.M.); 6Centro de Investigación Biomédica en Red Enfermedades Cardiovasculares (CIBERCV), 29010 Madrid, Spain; 7Department of Cardiology, Fundació Institut d’Investigació Sanitària Illes Balears (IdISBa), 07120 Palma, Spain; 8Department of Orthopedic Surgery and Traumatology, Hospital de la Santa Creu i Sant Pau, 08025 Barcelona, Spain; agonzalezo@santpau.cat; 9Department of Biochemistry, Hospital de la Santa Creu i Sant Pau, 08025 Barcelona, Spain; agarciao@santpau.cat (Á.G.-O.); jordonez1952@gmail.com (J.O.-L.); 10Department of Cardiology, Hospital Universitario Puerta de Hierro Majadahonda, 28222 Majadahonda, Spain; 11Geriatric Unit, Department of Internal Medicine, Hospital de la Santa Creu i Sant Pau, 08025 Barcelona, Spain; mbarcelot@santpau.cat; 12Cardiovascular Epidemiology Unit, Department of Cardiology, Hospital Universitari Vall d’Hebron, 08035 Barcelona, Spain; 13Foundation for Clinical Biochemistry & Molecular Pathology, 08025 Barcelona, Spain; 14CIBER Epidemiología y Salud Pública (CIBERESP), 28034 Madrid, Spain

**Keywords:** perioperative myocardial injury, screening, high sensitivity cardiac troponin T, noncardiac surgery

## Abstract

Perioperative myocardial injury (PMI) is a common cardiac complication. Recent guidelines recommend its systematic screening using high-sensitivity cardiac troponin (hs-cTn). However, there is limited evidence of local screening programs. We conducted a prospective, single-center study aimed at assessing the feasibility and outcomes of implementing systematic PMI screening. Hs-cTn concentrations were measured before and after surgery. PMI was defined as a postoperative hs-cTnT of ≥14 ng/L, exceeding the preoperative value by 50%. All patients were followed-up during the hospitalization, at one month and one year after surgery. The primary outcome was the incidence of death and major cardiovascular and cerebrovascular events (MACCE). The secondary outcomes focused on the individual components of MACCE. We included two-thirds of all eligible high-risk patients and achieved almost complete compliance with follow-ups. The prevalence of PMI was 15.7%, suggesting a higher presence of cardiovascular (CV) antecedents, increased perioperative CV complications, and higher preoperative hs-cTnT values. The all-cause death rate was 1.7% in the first month, increasing up to 11.2% at one year. The incidence of MACCE was 9.5% and 8.6% at the same time points. Given the observed elevated frequencies of PMI and MACCE, implementing systematic PMI screening is recommendable, particularly in patients with increased cardiovascular risk. However, it is important to acknowledge that achieving optimal screening implementation comes with various challenges and complexities.

## 1. Introduction

Perioperative myocardial injury (PMI) in noncardiac surgery is a frequent cardiac complication strongly associated with an increased rate of death and morbidities [1,2]. Eight million patients will experience a PMI, and one million will die within 30 days after surgery out of the 300 million yearly surgical procedures performed worldwide [3,4]. PMI is often missed because it can be clinically silent in more than 80% of cases; some PMI-alerting symptoms, like chest pain, are often masked or suppressed by postoperative sedation and analgesia [2,5]. Therefore, it is recommended to systematically measure a cardiac-specific biomarker, such as troponin (cTn), to facilitate the timely detection of PMI, particularly in patients with high cardiovascular risk [1,2,3,4,5]. Existing evidence emphasizes the importance of measuring cTn levels both before and after surgery, as this allows differentiation between acute cTn concentration increases associated with PMI and chronic elevations caused by other factors [1,5]. Moreover, the use of a high-sensitivity cTn (hs-cTn) assay enhances the early and sensitive detection of even minor instances of PMI [6,7].

In recent years, several national and international guidelines [8,9,10,11,12,13,14] have issued recommendations for the systematic screening of PMI using hs-cTn, instead of relying only on clinical symptoms. Despite the demonstrated efficiency and cost-effectiveness of systematic PMI screening with hs-cTn [15], especially for high cardiovascular risk patients undergoing major surgeries [16], its implementation is still limited in clinical practice. Consequently, there is scarce evidence derived from local practices regarding the implementation of systematic hs-cTnT screening programs. This situation may be attributed to a lack of established diagnostic criteria, limited available data on management strategies, and the uncertainty surrounding the impact of screening on patients, all of them important outcomes in real-world scenarios [2,17,18].

Given the growing importance of the early detection of perioperative PMI, we conducted a prospective cohort study to evaluate the feasibility and impact of implementing systematic PMI screening, which included measuring hs-cTn levels before and after surgery in high-risk noncardiac surgery patients. In this work, we present the clinical results and discuss the challenges and complexities encountered during the implementation process of the screening program at our institution.

## 2. Materials and Methods

### 2.1. Study Characteristics

#### 2.1.1. Study Site and Design

Our study was conducted at a single university tertiary hospital. It was an observational prospective study aimed at implementing systematic hs-cTnT screening in high-risk patients who underwent scheduled or urgent noncardiac surgery [19]. The choice of the biomarker to be measured was based on bibliographic antecedents [3,4,5,6,7] and on the previous experience of our researchers, who had participated in previous studies that used troponins as cardiac biomarkers [3,20]. Additionally, the criteria for conducting serial analysis and identifying abnormal values and patterns were drawn from studies evaluating hs-cTnT in daily practice and recommendations regarding its use for detecting myocardial injury [21,22,23,24]. Our hospital started measuring hs-cTnT, although as clinical practice tool (not for systematic screening) already in November 2009.

#### 2.1.2. Time of the Study

Our study was conducted between July 2016 and March 2019 and was registered at Clinicaltrials.gov (NCT03438448). We followed the STROBE reporting guidelines to ensure transparent reporting of our findings (Appendix A).

#### 2.1.3. Ethics and Patient Recruitment

Ethical approval for the study was provided by the Ethical Committee of Clinical Investigation at Hospital de la Santa Creu i Sant Pau, Barcelona (Spain). The approval, identified by the unique Protocol ID: IIBSP-IMP-2015-95, was issued on 11 May 2016. The study was conducted in accordance with the principles set forth in the Declaration of Helsinki.

The identification of eligible patients was carried out during the preoperative assessments by members of the research team. Prior to enrolment, all potentially eligible patients were informed about the study protocol and provided with detailed information. They voluntarily signed written informed consent, indicating their willingness to participate in the study after surgery and before hospital discharge.

#### 2.1.4. The Research Team and hs-cTnT Screening Implementation Management

Our study protocol was designed and accomplished by a newly formed, multidisciplinary research team that included investigators from the departments of Anesthesiology, Cardiology, Clinical Biochemistry, Clinical Epidemiology, and Surgery. They all worked together to supervise the execution of the protocol, evaluate the results, and identify any barriers that impeded its implementation.

#### 2.1.5. Data Collection and Management

During the hospitalization period, our research team performed daily reviews of the clinical records and charts of all enrolled patients until their discharge. This systematic process enabled the comprehensive and real-time collection of information pertaining to the patient’s medical condition and progress. In the one-month and one-year follow-up evaluations, patients or their family members were interviewed by phone to obtain information regarding the occurrence of the main outcomes of interest. If any such events were reported, the research personnel obtained pertinent source documents to validate and document the specific details. To facilitate data collection and management, an electronic case report form (eCRF) was created. This eCRF was designed in a secure online database (www.clinapsis.com), accessed on 1 May 2016, ensuring the confidentiality and integrity of the collected data.

### 2.2. Study Population

We included high-risk cardiovascular patients in our study. Inclusion criteria comprised aged ≥ 65 years or, if younger, had at least one documented antecedent of cardiovascular diseases (e.g., coronary artery disease, chronic heart failure, stroke, transient ischemic attack, or peripheral vascular disease) or impaired renal function (estimated glomerular filtration rate < 60 mL/min/1.73 m^2^). These patients underwent elective or urgent major noncardiac surgeries, including orthopedic, traumatological, spinal, visceral, digestive, peripheral vascular, thoracic, gynecologic, plastic, or otorhinolaryngologic procedures, and required at least an overnight hospital stay. We excluded patients aged < 65 years without cardiovascular diseases, those undergoing minor surgeries, and those operated on during weekends and holidays. Patients unable to comprehend the protocol and/or sign the informed consent were also excluded.

#### 2.2.1. Cardiac Biomarker Measurements

Hs-cTnT measurements were performed at three specific time points: preoperatively (at the preoperative visit or just before surgery) and at 48 and 72 h after surgery. For elective surgeries, the preoperative hs-cTnT measurements were requested by the surgeons and/or anesthesiologists conducting the preoperative visits. In the case of urgent, non-elective surgeries, the measurements were either requested by surgeons or anesthesiologists in charge or determined from plasma aliquots stored at 4 °C in the clinical laboratory for not longer than 24 h. The postoperative hs-cTnT measurements were requested by the attending physicians at recovery units, with support from the Clinical Epidemiology department. Hs-cTnT was measured by an electrochemiluminescent immune assay (Roche Diagnostics, Basel, Switzerland) with a measuring range of 5.0–10,000 ng/L, a limit of detection 5.0 ng/L, a 99th upper reference percentile (URL) of 14 ng/L, and 13 ng/L at the 10% coefficient of variation.

#### 2.2.2. PMI Definition, Diagnostic, and Management

The combination of pre- and postoperative hs-cTnT measurements was employed to differentiate between acute and chronic hs-cTnT elevations and to identify any pre-existing myocardial injury related to the surgery at an earlier stage [1]. The criteria for PMI were established based on a bibliography [25] and considering the prior experience of our research team. PMI was defined as a postoperative hs-cTnT concentration of ≥14 ng/L, along with an increase of ≥50% from the preoperative value. For myocardial infarction (MI), we followed the guidelines provided by the third universal definition [26], as our protocol was developed in the year 2016.

When PMI was identified, a formal cardiology evaluation was conducted, which included a thorough examination of all available clinical records, a 12-lead electrocardiogram (ECG), and an echocardiogram to detect any new regional wall motion abnormalities. PMI was ruled out when the elevated levels of hs-cTnT were determined by the cardiologists to be caused by noncardiac and/or non-ischemic factors such as sepsis, pulmonary embolism, or electrical cardioversion. Cardiologists, in collaboration with attending physicians, reviewed and assessed the use of medications such as aspirin, other antiplatelets, angiotensin-converting enzyme inhibitors (ACEI)s, statins, beta-blockers, and oral anticoagulants. Additionally, they evaluated the necessity for further diagnostic procedures, such as coronary angiography, and determined the potential need for coronary revascularization if deemed appropriate.

#### 2.2.3. Follow-Up

All the included patients underwent follow-up assessments during their initial hospitalization period, as well as one month and one year after surgery. We considered the assessment conducted within the first three days following surgery as the immediate postoperative assessment. Following the discharge, a personalized follow-up plan was recommended for some PMI patients based on their individual characteristics, involving either the hospital or primary care specialists responsible for the patients’ ongoing care. The discharge report included further details regarding the diagnosis of PMI (or MI) and provided recommendations for their appropriate management.

The follow-up evaluations at one month and one year following the initial surgery were performed by the research personnel. These assessments involved conducting phone interviews with the patients directly or with their close relatives. Additionally, electronic clinical records were reviewed to collect relevant information regarding outcome events and their associated specifics. The detailed descriptions of our screening program are shown in Figure 1.

### 2.3. Outcomes

Our primary outcome was to assess the incidence of death and major cardiovascular and cerebrovascular events (MACCE) within one month and one year after surgery. All-cause death was defined as any death attributed to a clearly documented non-vascular cause, while cardiovascular death was defined as any death attributed to cardiovascular or unknown causes. MACCE was defined as the composite outcome of major adverse cardiac and cerebrovascular events: myocardial infarction, congestive heart failure, new clinically relevant atrial fibrillation, stroke (including transient ischemic attack (TIA)), pulmonary embolism, and the need for cardiac revascularization. The definitions of these individual components were based on the guidelines in effect during the study period. The secondary outcomes were focused on the individual components of MACCE to allow for the analysis and understanding of the impact of each cardiovascular event on the overall outcome.

### 2.4. Statistical Analysis

Variables were presented as the percentage and number of cases for categorical variables or mean and standard deviation for quantitative ones; exceptions to this presentation are specifically mentioned in the Tables. Inferential statistics were employed to analyze the data and determine statistical significance. The prevalence and/or incidence of outcomes were reported along with their corresponding 95% confidence intervals. To assess the univariate associations between independent variables and the primary outcomes (death and MACCE), we employed various statistical tests such as chi-square test, Fisher’s exact test, *t*-tests, or non-parametric tests depending on the nature of the variable.

## 3. Results

During the study period, a total of 2333 eligible patients underwent surgery at our hospital. Out of these, 568 patients (24.3%) were excluded as they underwent surgery on weekends or holidays; 187 patients (8.01%) were not included due to a delay in their identification. Sixty-eight (2.91%) patients declined to participate, while 33 (1.41%) patients who initially agreed to participate later requested to withdraw from the study. Throughout the postoperative follow-up, which extended until March 2020, only one of the included surviving patients was lost to follow-up. The recruitment flowchart is shown in Figure 2.

### 3.1. Hs-cTnT Screening Implementation Management

Our multidisciplinary research team worked together to implement hs-cTnT screening in the tertiary hospital, showing the local experience as a real example for other sites. The information regarding our ongoing study has been effectively disseminated among the participating surgical departments in our center through informative meetings. Health professionals have shown great acceptance of the study protocol, even though we did not gather feedback through surveys or interviews. We have successfully integrated the request for preoperative hs-cTnT measurements into the anesthetic assessment templates for high-risk patients. Regrettably, due to the lack of support from the IT Department, we were compelled to manually request blood samplings, formal cardiology evaluations, and follow-up visit reminders. However, this manual approach proved to be insufficient for implementing systematic hs-cTnT screening into our routine clinical practice.

#### 3.1.1. Cardiac Biomarker Availability at Recruitment

In elective surgeries, preoperative hs-cTnT measurements were predominantly obtained during the month preceding the scheduled surgery, coinciding with the time of the decision for the intervention (68% of cases). In the remaining cases, preoperative samples for hs-cTnT were collected within the six months prior to surgery (30% of cases), and in a small percentage of cases (2% of cases), the preoperative samples were collected more than six months before the surgery. In contrast, for urgent surgeries, preoperative hs-cTnT measurements were obtained just a few minutes before the surgery.

#### 3.1.2. PMI Incidence, Clinical Data of Patients

In our cohort, PMI was observed in 232 patients, indicating a prevalence of 15.7% (95% CI 13.9–17.6) among the 1477 patients included in the study. The baseline characteristics of the 1477 patients are presented in Table 1, both as a group and stratified based on the presence or absence of PMI. No significant differences were observed in the occurrence of PMI based on age or sex. Most patients (94.8%) were included in the study due to being aged 65 years or older, and slightly over half of them were women (57.2%).

Compared to patients without PMI, those with PMI had a higher prevalence of preoperative comorbidities, which included: previous myocardial infarction (17.7% vs. 11.6%; *p* = 0.014), congestive heart failure (15.1% vs. 6.4%; *p* < 0.001), atrial fibrillation (24.1% vs. 13.4%; *p* < 0.001), stroke/transient ischemic attack (14.6% vs. 8.4%; *p* = 0.005), arterial hypertension (80.6% vs. 69.2%; *p* < 0.001), diabetes mellitus (31.0% vs. 23.5%; *p* = 0.017), and impaired renal function (31.2% vs. 15.7%; *p* < 0.001). Consequently, PMI patients more frequently had higher Revised Cardiac Risk Indexes (III–IV) than those without PMI (17.7% vs. 10.3%). Additionally, PMI patients also presented significantly worse values of certain baseline laboratory variables, such as lower estimated glomerular filtration rate (*p* < 0.001) and hemoglobin concentration (*p* < 0.001) and higher preoperative hs-cTnT value (*p* < 0.001).

#### 3.1.3. Immediate Postoperative Assessment

A total of 155 (10.5%) postoperative hs-cTnT measurements were not recorded in patients who were discharged before the three-day mark after the intervention or during weekends. This was attributed to attending physicians being unaware of the study protocol. Similarly, as in the case of preoperative values, hs-cTnT concentrations were observed to be significantly higher (2 to 2.5-fold) in patients with PMI compared to patients without PMI at both 48 and 72 h after surgery (*p* < 0.001 for both comparisons). Among the 232 patients with PMI, a postoperative ECG was obtained in 151 patients (65.1%), and formal cardiology consultation was conducted in 132 patients (56.9%). Additionally, all 132 patients who received cardiology consultation underwent an additional echocardiogram. Most PMI patients in our cohort (220; 94.8%) were asymptomatic.

New or presumed new changes in ECG were observed in 6.0% (9 out of 151) of the postoperative ECGs, and presumed new echocardiographic wall motion abnormalities in 11.4% (15 out of 132) of the performed echocardiographic tests. Thus, only 3.0% (7 out of 232) of the PMI patients met the criteria for myocardial infarction according to the existing international definitions. PMI patients experienced a higher incidence of hemodynamic instability and bleeding compared to non-PMI patients. The most notable differences included a higher occurrence of significant hypotension (37.3% vs. 29.0%, *p* = 0.014) requiring vasopressor drugs (26.1% vs. 15.9%; *p* = 0.026), significant tachycardia (30.0% vs. 18.3%; *p* < 0.001), immediate postoperative bleeding (17.3% vs. 8.1%; *p* < 0.001), and postoperative shock (6.3% vs. 2.5%; *p* = 0.008) (Table 2).

### 3.2. Follow-Up and Outcomes

We achieved a significantly high level of compliance with telephone follow-ups conducted by our research team. The compliance rate was 99.9% at both the one-month and one-year marks. As shown in Figure 3 (Appendix A), in the whole cohort, infection was the most common outcome at one month after surgery, occurring in 15.9% of patients. However, the rate of infection decreased significantly in the subsequent months (7.7%, *p* = 0.02). On the other hand, all the MACCE and its individual components, except for new atrial fibrillation or myocardial infarction, were more frequently observed in the months following the first month after surgery. During the first month, a total of 25 all-cause deaths occurred, accounting for 1.7% (95% CI: 1.1–2.4%) of the entire cohort. In the following months, there was a significant increase in all-cause deaths, with a total of 104 deaths (7.0% of the survivors, *p* < 0.001 compared to the first month). Among these deaths, those attributed to cardiovascular causes also increased over time. The rate of cardiovascular deaths in the whole cohort was 0.9% in the first month and increased to 2.1% among survivors in the subsequent months (*p* = 0.010).

At the one-month follow-up, we observed that PMI patients had significantly higher rates of infection (23.3% vs. 14.5%, *p* = 0.001), all MACCE (9.5% vs. 3.6%, *p* < 0.001), and myocardial infarction (3.0% vs. 0.3%, *p* < 0.001) compared to patients without PMI. These outcomes (infection, all MACCE, and MI) were the only ones that exhibited significant differences between the two groups of PMI and no-PMI patients. The all-cause death rate in PMI patients was 1.7% in the first month after surgery, which increased to 11.2% at one year. It is important to note that at one month, no significant differences were observed between patients with PMI and those without PMI in terms of all-cause or cardiovascular deaths. However, in the subsequent months up to one year, these differences became more notable with a higher percentage of deaths in the PMI group (11.2% vs.6.3%, *p* < 0.001). Additionally, the occurrence of congestive heart failure (6.0% vs. 2.6%, *p* = 0.001) and the need for coronary revascularization (1.4% vs. 0%, *p* = 0.001) were significantly higher in the PMI group. This indicates a distinct pattern of increasing cardiovascular outcomes beyond the initial month following surgery. Taken together, these results indicate that PMI patients experienced a high incidence of MACCE throughout the entire follow-up period. During the first month follow-up, both MI and congestive heart failure (CHF) occurred with similar frequency in PMI patients. However, in the subsequent months, there was a predominance of CHF as compared to MI. This suggests that the occurrence of CHF becomes more prominent as time progresses following PMI.

### 3.3. Cardiovascular Drugs Use Pre and Postintervention

In our cohort, cardioactive drugs were more frequently used in PMI patients compared to those without PMI, both before and after surgery, except for statins and antiplatelets. The percentage of PMI patients receiving aspirin (AAS), other antiplatelets, beta-blockers, ACEIs, and oral anticoagulants was significantly higher than in no-PMI patients at baseline (*p*-values ranging from 0.003 to 0.009), at the first month after the intervention (*p*-values ranging from 0.032 to 0.001), and during the period between the first and subsequent months of the postoperative period (*p*-values ranging from 0.006 to 0.001). Among these drugs, angiotensin-converting enzyme inhibitors were the most prescribed. Importantly, the percentage of prescribed drugs did not differ significantly between PMI and no-PMI patients throughout the postoperative period (Appendix A).

## 4. Discussion

Recent international guidelines recommend [14] or suggest [27] implementing systematic PMI screening with hs-cTnT in high-risk patients undergoing noncardiac surgery. However, there is still limited experience of such implementation in local practice [2,17,18]. Our cohort study proves the effectiveness of systematic hs-cTnT screening for high-risk patients at a university hospital, revealing a high PMI prevalence, especially in patients with a history of cardiovascular risk factors (CVRF) or cardiovascular disease (CVD). PMI was associated with intraoperative or immediate postoperative complications such as hypotension, bleeding, or arrhythmias. Furthermore, during the follow-up period, PMI patients had higher rates of mortality and major cardiovascular adverse events. These observations pointed out the need for and usefulness of implementing systematic PMI screenings in high-risk patients and aligned with recent ESC guidelines [14], despite being issued five years after our study began. This accomplishment stems from the expertise and prior experience of our research team, actively participating in generating evidence on the study topic [3,20,22,23]. Although developed in 2016, our approach remains in line with ESAIC’s more cautious recommendations in 2023 [27].

### 4.1. Screening Implementation. Barriers and Facilitators

Despite some challenges and barriers, our dedicated research team successfully recruited two-thirds of all eligible high-risk patients and followed them for up to one year. The exclusion of weekend and holiday surgeries, primarily due to personnel unavailability, resulted in the loss of 24.3% of eligible cases. For the same reason, an additional 8.01% of cases were lost by delayed identification. In contrast, a minimal number of patients declined to participate, either before or after inclusion. Despite lacking the IT Department’s assistance to ensure automated requests, we were able to obtain extensive clinical data during pre, peri, and immediate postoperative steps. We achieved successful one-month and one-year follow-ups for all but one participant.

Analyzing barriers and facilitators, we found logistical issues, such as personnel unavailability or lack of IT support as the main barrier, that exceeded the competence of the investigators, causing the data loss. In contrast, the patients’ easy understanding of the study protocol and the dedication of the research team during the recruitment and follow-up processes served as facilitators for our study, as both are well-known contributors to the successful achievement of implementation objectives, such as those related to our screening program [28].

### 4.2. PMI Diagnosis, Incidence, and Characteristics

We defined PMI as hs-cTnT elevation above the 99th upper reference percentile, with a change of more than 50% from preoperative values. These criteria were established based on available information and our clinical experience [25,26,29,30]. Pre and postoperative hs-cTnT concentrations allowed us to differentiate its acute changes from chronic elevations. Our criteria differed from some previous studies that only focused on postoperative hs-cTnT measurements [3,20], used contemporary (i.e., not high sensitivity) cTnT methods [20], or measured high-sensitivity cardiac troponin I (hs-cTnI) [31]. Most of the preoperative samples were obtained in scheduled, non-urgent surgeries during the month before the intervention, while few others were obtained up to 6 months prior. A large intraindividual variability of hs-cTnT over time could be a confounding factor for PMI diagnosis when comparing pre- and postoperative values. Nevertheless, hs-cTnT concentrations have low long-term variability, being 11% in healthy individuals and 8.5% in patients with CVRF or stable coronary disease, like those included in study [32]. Since we used a criterion for PMI diagnosis of a pre to postoperative hs-cTnT variation of ≥50%, much higher than the long-term biological variability, we inferred that even samples obtained 6 months before surgery accurately reflected the true troponin value at the time of the intervention.

In our study, PMI was diagnosed in one in six (15.7%) included patients. Despite variations in criteria and observational periods, the elevated PMI incidence in our study concurs with findings from other comparable studies using hs-cTnT as a biomarker, which found a PMI prevalence of 17.9% [1]. Patients with PMI exhibited a higher prevalence of antecedents of CVRF and higher preoperative hs-cTnT concentrations. It was not surprising that PMI patients had significantly higher preoperative hs-cTnT values than patients without PMI, as hs-cTnT is a sensitive biomarker of the cardiac stress associated with pre-existing CVRF and CVD. Furthermore, PMI patients experienced more frequent cardiocirculatory complications during or immediately after surgery. Many of the observed complications, such as hypotension, bleeding, or arrhythmia, can disbalance the equilibrium between cardiac oxygen supply and demand and lead to myocardial injury unrelated to atherothrombosis [33], causing type 2 myocardial infarction. These observations suggested that among our high-risk patients, some were more prone to PMI than others. Elevated hs-cTnT concentrations before surgery, along with the presence of CVRF or CVD, delineated what could be denominated as a “PMI-prone phenotype”. Recognizing this phenotype could aid clinicians in providing special attention to these patients during peri- and postoperative periods, as well as during follow-ups outside the hospital. Additionally, PMI was more often observed in patients undergoing urgent surgeries, leading to the hypothesis that the lack of adequate hemodynamic stability or control of pre-existing CV conditions, combined with the stress imposed by the urgent surgical condition, could lead to PMI. Overall, our findings emphasize the importance of systematic PMI screening to identify patients prone to PMI and optimize their health and functional capacity prior to surgery to improve their surgical outcomes [34].

### 4.3. Outcomes in the Follow-Ups

In the whole cohort, the incidence of MACCE was 4.5%, reaching 9.5% in PMI patients in the first month. A comparable study reported MACCE incidences of 9.0%, although it included patients exclusively undergoing vascular surgery [35]. Another Spanish cohort study observed the incidence of MACCE of 4.3% in intermediate and high-risk surgeries, although the study included fewer patients than ours, with higher stages of cardiac risk or undergoing urgent surgeries and a follow-up period limited to hospitalization [36].

PMI patients had significantly higher rates of MACCE in the first month (9.5% vs. 3.6%) and successive months (8.6% vs. 4.3%) than no-PMI ones. All-cause and cardiovascular death had the same incidence (1.7% and 0.9%, respectively, in both groups) in PMI and no-PMI patients during the first month follow-up. However, at the one-year observation, both all-cause (11.2%) and cardiovascular (3.9%) deaths had sharply increased in PMI patients, outlining the prognostic severity of the condition and emphasizing the importance of systematic screening. Notably, during the months after the first follow-up, PMI showed a high rate of CHF (6.0% of cases). This fact reinforces the importance of systematically detecting PMI during hospitalization, as well as monitoring signs or symptoms of CHF during follow-up. It is suggestive to hypothesize that optimal management of risk factors associated with CHF, such as hypertension, diabetes, obesity, and smoking, can help reduce its occurrence in PMI patients. It was not surprising that some of the PMI patients (1.4%) required coronary revascularization during the one-year follow-up compared with no case in no-PMI patients. This finding aligns with a published meta-analysis [37], which included coronary revascularization trials and defined PMI based on larger biomarker elevations associated with subsequent mortality. In the case of other complications unrelated to the CV system, infection was predominant in the first month (15.9%), but its incidence declined in successive months to 7.7%, which was equal to the incidence of all-cause death (7.0%).

Our cohort included old patients with a high CVRF; they were commonly using cardiovascular drugs. Preoperatively, the use of ACEIs, aspirin, beta-blockers, or oral anticoagulants, ordered by frequency of use, was significantly more prevalent in patients who later experienced PMI. This differential therapeutic pattern remained consistent throughout the entire follow-up period. Statins were widely and similarly used in PMI and no-PMI patients. Thus, the diagnosis of a PMI did not result in a change after surgery in the frequency of drug usage, which could be attributed to the lack of compelling evidence regarding specific treatments for PMI patients. We cannot consider the MANAGE trial [38] launched after our study’s beginning, but clinicians implicated in our study relied on the 2014 ESC/EAS (European Atherosclerosis Society) guideline [8], which recommended considering the use of aspirin and statins based on patients’ cardiovascular risk factors. Moreover, given that factors observed as PMI-associated pointed to a coronary oxygen disbalance rather than to an atherothrombotic episode as the cause of the myocardial injury, the lack of increased use of aspirin, or anticoagulant/antiplatelet drugs after PMI, as we observed in our study, seemed appropriate.

### 4.4. Implications for Clinical Practice

The current real experience in implementing hs-cTnT screening can serve as a compelling example for other sites. Our PMI screening yielded significant results in high-risk patients, especially those with a history of CVRF, CVD, or renal impairment. The screening is feasible as it only requires clinical surveillance of patients and routine clinical and laboratory explorations. Identifying PMI patients opens the opportunity to enhance their outcomes. Of note, we identified certain implementation barriers that need to be addressed for the optimal success of the screening program.

### 4.5. Limitations and Strengths

The main limitation of the study was the incomplete recruitment of all eligible patients. Although we included two-thirds of the eligible patients, the lack of sufficient investigators to conduct assessments during weekends, holidays, or in a timely manner according to the protocol was a logistical barrier to achieving complete recruitment. The absence of centralized technical assistance through staff departments, such as IT, resulted in a minor limitation in the availability of all registrable data. Tasks like requesting blood samplings, conducting formal cardiology evaluations, and sending follow-up visit reminders were performed manually. However, the investigators were able to overcome this barrier through intense dedication. These implementation barriers could be easily overcome if executive boards or key decision-makers provide support and allocate the necessary resources for the implementation process. We have some limitations in comparing our results with other studies due to the absence of standardized criteria for PMI diagnosis. However, our study design was planned in anticipation and alignment with the most recent guidelines on the topic. Finally, due to the experimental nature of our study, it was solely conducted by the group of researchers involved. Thus, our efforts did not provide enough grounds to incorporate hs-cTnT systematic screening into clinical practice.

The study also has some strengths. The study was developed by an ad hoc established multidisciplinary research team comprising different clinical professionals. This collaborative approach significantly improved the applicability and compliance of our protocol. The study also showed the challenges and complexities that exist in real clinical practice when implementing a local PMI screening program. It highlights that the mere publication and dissemination of guidelines and recommendations do not automatically ensure their rigorous implementation in clinical practice. Factors such as awareness, acceptance, and understanding among healthcare professionals, as well as the availability of necessary resources and infrastructure, are crucial for the successful implementation of new protocols. Finally, an easy understanding of the study protocol and the dedication of the research team during the recruitment and follow-up processes allowed for almost complete compliance of patients in both the one-month and one-year follow-ups.

## 5. Conclusions

Our study demonstrated the relevance of implementing PMI screening and supported the need for its systematic screening in high-risk patients undergoing elective or urgent noncardiac surgeries. The study also revealed that challenges and complexities exist in real clinical practice when implementing a local PMI screening program. The successful implementation of new protocols in clinical practice will rely not only on the implication of healthcare professionals but also on the availability of necessary resources and infrastructure.

## 6. Patients

Patients were not involved in the design, conduct, or reporting of this study as it was not applicable to this research project.

## Figures and Tables

**Figure 1 jcm-12-05371-f001:**
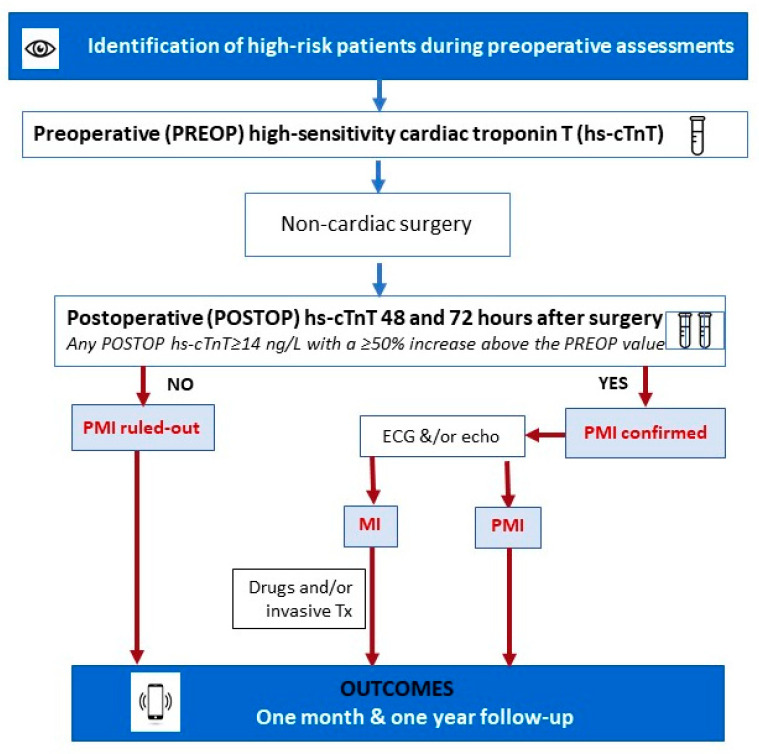
Systematic PMI screening with hs-cTnT.

**Figure 2 jcm-12-05371-f002:**
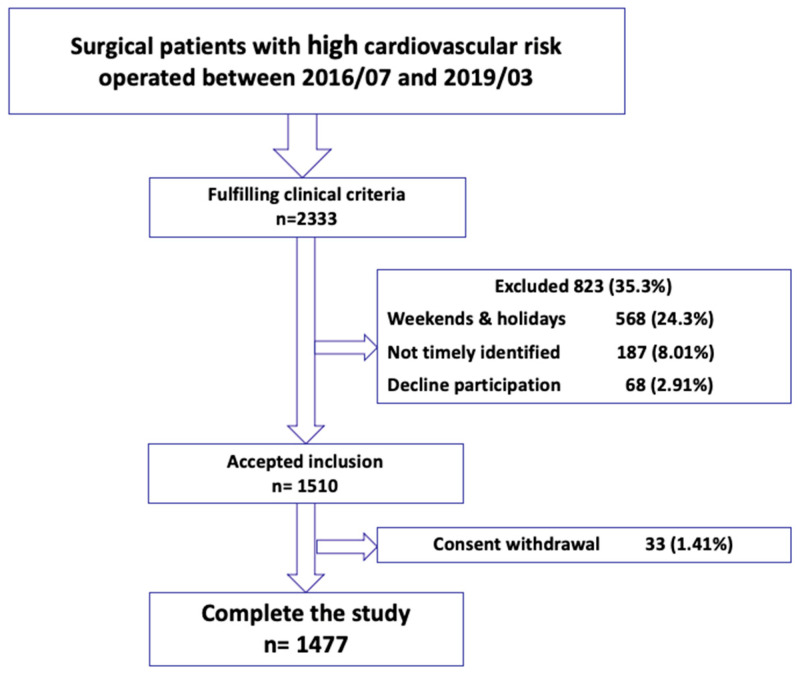
Recruitment flowchart.

**Figure 3 jcm-12-05371-f003:**
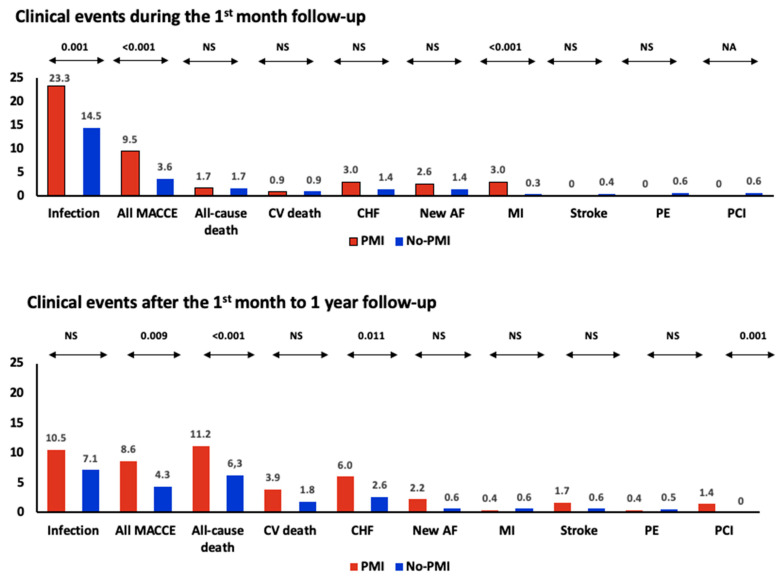
Percentages of clinical events observed in the earlier (first month, 1st-m) and subsequent (first month to one year, 1 y) follow-up periods. p-values are for comparison between respective follow-up periods in each subgroup (PMI, No-PMI) of patients. Abbreviations: NS—Not significant: MACCE—Major adverse cardiovascular and cerebrovascular events; CHF—congestive heart failure; AF—atrial fibrillation; MI—myocardial infarction; PE—pulmonary embolism; PCI—percutaneous coronary intervention. Definitions: All-cause death—any death attributed to a clearly documented non-vascular cause. Cardiovascular death—any death attributed to cardiovascular or unknown causes.

**Table 1 jcm-12-05371-t001:** Baseline characteristics of the study population according to the presence (PMI) or absence (No PMI) of perioperative myocardial injury after surgery.

Baseline Characteristics	All Patients % (*N* 1477)	PMI % (*N* 232)	No PMI % (*N* 1245)	*p*-Value
Age ≥ 65 years	94.8 (1399)	96.1 (223)	94.5 (1176)	NS
Women	57.2 (843)	58.2 (135)	57.1 (708)	NS
**Antecedents at the preoperative visit**				
Myocardial infarction	12.5 (185)	17.7 (41)	11.6 (144)	0.014
Congestive heart failure	7.7 (114)	15.1 (35)	6.4 (79)	<0.001
Atrial fibrillation	15.1 (223)	24.1 (56)	13.4 (167)	<0.001
Stroke/TIA	9.4 (139)	14.6 (34)	8.4 (80)	0.005
Pulmonary embolism	1.4 (21)	0.4 (1)	1.6 (20)	NS
Deep vein thrombosis	2.0 (30)	2.2 (5)	2.0 (25)	NS
Peripheral artery disease	10.4 (153)	13.8 (32)	9.7 (121)	NS
Arterial hypertension	71.0 (1048)	80.6 (187)	69.2 (861)	<0.001
Diabetes mellitus	24.7 (364)	31.0 (72)	23.5 (292)	0.017
Dyslipidemia	50.9 (751)	55.2 (128)	50.1 (623)	NS
COPD	13.6 (200)	11.6 (27)	13.9 (173)	NS
Impaired renal function	18.1 (267)	31.2 (72)	15.7 (195)	<0.001
Revised Cardiac Risk Lee Index				
I	56.2 (828)	47.2 (109)	57.9 (719)	0.003
II	32.4 (477)	35.1 (81)	31.9 (396)	
III	8.1 (120)	13.4 (31)	7.2 (89)	
IV	3.3 (48)	4.3 (10)	3.1 (38)	
eGFR (mL/min/1.73 m^2^)				
≤30	5.0 (72)	7.1 (16)	4.6 (56)	<0.001
31–59	21.0 (305)	31.6 (71)	19.1 (234)	
≥60	74.0 (1075)	61.3 (138)	76.4 (937)	
Preoperative hemoglobin (g/L)				
≤100	10.7 (157)	16.5 (38)	9.6 (119)	<0.001
101–129	38.9 (572)	44.2 (102)	37.9 (470)	
≥130	50.5 (743)	39.4 (91)	52.5 (652)	
Preoperative hs-cTnT (ng/L) *	13 (9–22)	15 (11–26)	13 (9–21)	<0.001

* Hs-cTnT as median (interquartile range). Abbreviations: NS—Not significant: PMI—perioperative myocardial injury; TIA—transient ischemic attack; COPD—chronic obstructive pulmonary disease; eGFR—estimated glomerular filtration rate (using the CKD-EPI formula); Hs-cTnT—cardiac troponin T measured with a high-sensitivity method.

**Table 2 jcm-12-05371-t002:** Surgery characteristics and intra- and immediate postoperative clinical complications.

	All Patients % (*N* 1477)	PMI % (*N* 232)	No PMI % (*N* 1245)	*p*-Value
Priority of surgery				
Elective	71.6 (1054)	58.9 (136)	74.0 (918)	<0.001
Urgent	28.4 (418)	41.1 (95)	26.0 (323)	
	**INTRAOPERATIVE Complications**			***p*-Value**
Significant arterial hypotension	71.8 (1043)	77.2 (176)	70.8 (867)	0.044
Requiring treatment	59.0 (619)	70.3 (123)	56.8 (496)	0.001
Significant arterial hypertension	43.5 (637)	44.8 (103)	43.2 (534)	NS
Requiring treatment	10.6 (67)	9.8 (10)	10.8 (57)	NS
Significant tachycardia	9.3 (135)	12.3 (28)	8.7 (107)	NS
Requiring treatment	8.2 (12)	16.7 (5)	6.0 (7)	NS
Significant bradycardia	32.2 (470)	32.3 (74)	32.2 (396)	NS
Requiring treatment	10.3 (48)	13.9 (10)	9.7 (38)	NS
Bleeding	4.0 (59)	7.8 (18)	3.3 (41)	0.004
Shock	3.9 (57)	8.2 (19)	3.1 (38)	0.001
Hypovolemic	96.4 (54)	94.4 (17)	97.4 (37)	NS
Distributive	3.6 (2)	5.6 (1)	2.6 (1)	NS
Significant hypoxemia (SaO_2_ < 90%)	2.0 (30)	2.6 (6)	1.9 (24)	NS
	**Immediate POSTOPERATIVE Complications** **(in the First 3 Postoperative Days)**			***p*-Value**
Significant arterial hypotension	30.3 (438)	37.3 (85)	29.0 (353)	0.014
Requiring treatment	17.8 (87)	26.1 (24)	15.9 (63)	0.026
Significant arterial hypertension	58.5 (848)	54.1 (124)	59.3 (724)	NS
Requiring treatment	7.0 (59)	8.1 (10)	6.8 (49)	NS
Significant tachycardia	20.0 (291)	30.0 (68)	18.3 (223)	<0.001
Requiring treatment	5.7 (17)	9.0 (6)	4.8 (11)	NS
Significant bradycardia	37.9 (547)	31.9 (73)	38.1 (474)	0.04
Requiring treatment	1.7 (9)	0.0 (0)	2.0 (9)	0.105
Bleeding	9.6 (139)	17.3 (40)	8.1 (99)	<0.001
Shock	3.1 (45)	6.3 (14)	2.5 (31)	0.008
Hypovolemic	93.5 (43)	100.0 (14)	90.6 (29)	NS
Distributive	4.3 (2)	0.0 (0)	6.3 (2)	NS
Septic	2.2 (1)	0.0 (0)	3.1 (1)	NS
Significant hypoxemia (SaO_2_ < 90%)	15.2 (218)	18.5 (41)	14.6 (177)	NS
Postoperative hs-cTnT (ng/L) *				
48 h	16 (10–27)	32 (20–56)	14 (9–22)	<0.001
72 h	14 (9–24)	30 (17–52)	13 (8–20)	<0.001

* Hs-cTnT as median (interquartile range). Abbreviations: NS—Not significant: PMI—perioperative myocardial injury; Hs-cTnT—cardiac troponin T measured with a high-sensitivity method. Definitions: Significant hypotension: drop of 30% of arterial systolic blood pressure (SBP) from baseline (before anesthetic induction); significant hypertension: SBP > 140 mmHg and/or diastolic blood pressure (DBP) > 90 mmHg; significant tachycardia: heart rate (HR) > 100 beats per minute (bpm); significant bradycardia: HR < 60 bpm; bleeding: postoperative hemoglobin (Hb) concentration < 70 g/L + need of transfusion of 2 pools or Hb drop of >50 g/L from preoperative value or transfusion of >3 (FBC) within 24 h; significant hypoxemia—oxygen saturation (SaO_2_) < 90%.

## Data Availability

The data used in the present study are part of a larger dataset. The datasets generated and analyzed during the current study are available and can be supplied by the corresponding authors upon reasonable request. The data not used for this manuscript will be employed in future manuscripts.

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
