# Peer review of "The Relevance of Implementing the Systematic Screening of Perioperative Myocardial Injury in Noncardiac Surgery Patients"

_jcm, 2023, doi:10.3390/jcm12165371_

Round 1

Reviewer 1 Report

Comments/suggestions.

Abstract:

1.     Lines 34-35: Kindly update the sentence.

2.     Line 37: consider using followed-up instead of follow-upped.

3.     Line 39: consider using MACE or add ‘cerebrovascular’ for the MACCE. 

4.     Line 41: may consider using ‘suggesting’ instead of ‘exhibting’.

5.     Line 42: may consider using ‘increased’ instead of ‘more’.

6.     Lines 44-45: may consider revision of the recommendation for clarity.

Introduction:

1.     Line 54-56: you may want to consider revising the sentence starting with PMI, as it gives the impression that chest pain is a typical symptom of PMI. 

2.     Line 57: consider ‘such as troponin’ instead of ‘as troponin’.

3.     Lines 61-62: consider ‘high-sensitivity cTn (hs-cTn) assay’ instead of ……‘measures (hs-cTn)’.

4.     Line 68: consider using ‘in’ in place of ‘into’.

5.     Lines 71-71: consider revising the last part of the sentence (………and the impact of screening…..) for clarity. 

Materials and Methods:

1.     Line 83: consider using aimed at instead of aimed to. 

2.     Lines 84-90: the sentence is rather too long. You may want to consider a revision using more than one sentence for clarity. 

3.     Lines 90: ‘furthermore’ may be deleted.

4.     Line 99: identity of the person that granted approval may not be necessary!

5.     Line 148: consider using anesthesiologist in charge, instead of at charge.

6.     Line 198: the suggestion on MACCE above.

7.     Lines 209-217 (statistical analysis): given the context of your study design, the statistical method employed may be revisited. The authors may want to consider using the Kaplan-Meier curve. Similarly, it may be worthwhile to correct for the confounding CVR in determining the role of PMI on account of interest. 

Results:

1.     Line 223: consider removing only.

2.     Lines 239-243: consider indicating most of these as limitations. 

3.     Line 256: consider removing whole and add a comma after ‘group’.

4.     Line 305: may consider using n (%) for consistency.

Discussion:

1.     Line 411: may consider revising the sentence….screening resulted in the detection of…….

2.     Lines 418-437: essentially discussing the limitations of the study ahead of a discussion of the main findings. The authors may want to revise this. 

3.     Line 441: consider frequent for frequently.

4.     Line 445: These observations provided evidence…..

5.     General comments on the discussion: 

·       The authors could have adequately and succinctly discussed the findings of the study. They should consider a revision and be chronological. 

·       The sub-headings are rather too many.

Conclusions:

1.     May consider revising to emphasize the findings of your study. 

References:

1.     Adequate and appropriate. 

I have included the comments and suggestions.

Author Response

Dear editors,

Thank you for the opportunity to resubmit a revised version of our manuscript. We have considered the comments of the reviewers and the editorial team and provided a point-to-point reply below with our comments in italics followed by questions in normal font. We also took opportunity and have corrected some orthographic and grammar spelling throughout the manuscript, as well as updated references order. The changes in the manuscript are highlighted as required.

Reviewer 1

We sincerely thank the reviewer for the interest in our study, providing constructive suggestions and important observations, which have allowed us to significantly improve the whole text, especially the Discussion section. We have now accordingly made substantive changes to various parts of the main text, especially at the Discussion and Conclusion sections.

Abstract:

  1. Lines 34-35: Kindly update the sentence.

Thank you for the suggestion. We have reworded the sentence to "We conducted a prospective, single-center study aimed at assessing the feasibility and outcomes of implementing systematic PMI screening."

  1. Line 37: consider using followed-up instead of follow-upped.

Updated. We apologize by this clerical error.

  1. Line 39: consider using MACE or add ‘cerebrovascular’ for the MACCE. 

Thank you for this important observation. We added “cerebrovascular” at MACCE definition, it now reads:  major cardiovascular and cerebrovascular events (MACCE).

  1. Line 41 (now line 42): may consider using ‘suggesting’ instead of ‘exhibiting’. Updated.
  2. Line 42 (now line 43): may consider using ‘increased’ instead of ‘more’. Updated.
  3. Lines 44-45 (now lines 45-48): may consider revision of the recommendation for clarity.

Thank you for the consideration, we updated the sentence to clarify it as: "Given the observed elevated frequencies of PMI and MACCE, implementing systematic PMI screening is recommendable, particularly in patients with increased cardiovascular risk. However, it's important to acknowledge that achieving optimal screening implementation comes with various challenges and complexities.”

Introduction:

  1. Line 54-56 (now lines 56-58): you may want to consider revising the sentence starting with PMI, as it gives the impression that chest pain is a typical symptom of PMI. 

Thank you for important observation. The updated sentence aims to improve its reading and comprehension: "PMI is often missed because it could be clinically silent in more than 80% of cases; some PMI-alerting symptoms, like chest pain, are often masked or suppressed by postoperative sedation and analgesia."

  1. Line 57 (now line 60): consider ‘such as troponin’ instead of ‘as troponin’. Updated.
  2. Lines 61-62 (now line 65): consider ‘high-sensitivity cTn (hs-cTn) assay’ instead of ……‘measures (hs-cTn)’. Updated.
  3. Line 68 (now line 70): consider using ‘in’ in place of ‘into’. Updated.
  4. Lines 71-71 (now lines 73-74): consider revising the last part of the sentence (………and the impact of screening…..) for clarity. 

Thanks for the suggestion; the sentence required improvement. We updated it and now it reads: "This situation may be attributed to lack of stablished diagnostic criteria, limited available data on management strategies, and the uncertainty surrounding the impact of screening on patients', all of them important outcomes in real-world scenarios."

Materials and Methods:

  1. Line 83 (now line 86): consider using aimed at instead of aimed to.  Updated.
  2. Lines 84-90 (now lines 87-93): the sentence is rather too long. You may want to consider a revision using more than one sentence for clarity. 

The reviewer is fully right. By breaking the sentence, the information becomes more organized and easier to understand. Updated version now reads: "The choice of the biomarker to be measured was based on bibliographic antecedents [3-7] and on the previous experience of our researchers, who had participated in previous studies that used troponins as cardiac biomarkers [3,20]. Additionally, the criteria for conducting serial analysis and identifying abnormal values and patterns were drawn from studies evaluating hs-cTnT in daily practice and recommendations regarding its use for detecting myocardial injury."

  1. Lines 90 (now line 93): ‘furthermore’ may be deleted. Updated.
  2. Line 99 (now line 102): identity of the person that granted approval may not be necessary! Updated.

Thank you for your observation. We would like to clarify that we included the identity as per the request of other journals where we have previously published studies requiring ethical approval by the corresponding committee. These journals specifically requested these details. However, attending the reviewer suggestion we have removed the identity of person at charge of the approval. If requested by the Editors, we will provide them with the information.

  1. Line 148 (now line 151): consider using anesthesiologist in charge, instead of at charge. Updated.
  2. Line 198 (now lines 201 and 206): the suggestion on MACCE above. Updated.
  3. Lines 209-217 (statistical analysis): given the context of your study design, the statistical method employed may be revisited. The authors may want to consider using the Kaplan-Meier curve. Similarly, it may be worthwhile to correct for the confounding CVR in determining the role of PMI on account of interest. 

Thank you for the interesting suggestions. To determine whether the Kaplan-Meier representation would be appropriate, we requested advice from our statistical advisor. Following his recommendation, we decided not to use it. The reason is that assessing survival would not hold much validity, given the inherent imprecision of the time estimates for evaluating of clinical events at only two specific follow-up time points (1-month and 1-year). As mentioned in the text, we have protocolized only two follow-up observations due to our personnel limitations.

Based on the advice provided by our statistical advisor, we acknowledge that several of the clinical events evaluated could be considered potential confounding factors related to CVR. However, we decided not to implement corrections for one or more of these variables, as doing so would lead to the creation of numerous models, and consequently, collinearity problems might arise frequently.

Results:

  1. Line 223 (now lines 227-228): consider removing only.

We have eliminated “Only” from the sentence and also took the opportunity to correct it grammatically, that now reads “Sixty-eight (2.91%) patients declined to participate, while 33 (1.41%) patients who initially agreed to participate later requested to withdraw from the study”.

  1. Lines 239-243: consider indicating most of these as limitations. 

Thank you for your observation. We updated the corresponding sentence at the limitation section (Lines: 539-543) “The absence of centralized technical assistance through staff departments such as IT resulted in a minor limitation in the availability of all registrable data. Tasks like requesting blood samplings, conducting formal cardiology evaluations, and sending follow-up visit reminders were performed manually.”

  1. Line 256 (now line 259): consider removing whole and add a comma after ‘group’. Updated.
  2. Line 305 (now line 309): may consider using n (%) for consistency. Updated.

Discussion:

  1. Line 411 (now lines 414-417): may consider revising the sentence….screening resulted in the detection of…….

We updated the sentence, considering your suggestion: Our cohort study proves systematic hs-cTnT screening effectiveness for high-risk patients at a university hospital, revealing a high PMI prevalence, especially in patients with a history of cardiovascular risk factors (CVRF) or cardiovascular disease (CVD).”

  1. Lines 418-437: essentially discussing the limitations of the study ahead of a discussion of the main findings. The authors may want to revise this. 

We used to refer the limitations and strengths of our studies at the end of the Discussion, as it is also usual in many papers describing limitations and strengths. Thus, respectfully we prefer to keep the previous order.

  1. Line 441 (now line 468): consider frequent for frequently. Updated.
  2. Line 445 (now lines 473-474): These observations provided evidence…..

We have updated the sentence, it now reads: “These observations suggested that among our high-risk patients some were more prone to PMI than the others.”

  1. General comments on the discussion: 
  • The authors could have adequately and succinctly discussed the findings of the study. They should consider a revision and be chronological. 

Thank you for this important observation. Following the advice, we have accordingly updated and re-organized the Discussion section maintaining its chronology. The updated section 4.2 (before section 4.1.3) has been revised, and it now begins with the second paragraph.

  • The sub-headings are rather too many.  

The reviewer is right, in response to this advice, we have reordered, improved, and reduced some subheadings, avoiding some not substantial information, which resulting in a 10% reduction in the Discussion text. The sections: “4.1.1 Implementation” and “4.1.2 Barriers and facilitators” were fashioned to one: “4.1. Screening Implementation. Barriers and facilitators”.  The section "4.2. Current Study in the Context of Previous Research" has been fused and integrated within the framework of the new subheadings.

Conclusions:

  1. May consider revising to emphasize the findings of your study. 

We have updated conclusion according to our updated Discussion section.

References:

  1. Adequate and appropriate. 

Comments on the Quality of English Language

I have included the comments and suggestions.

Submission Date

16 July 2023

Date of this review

23 Jul 2023 20:02:18

Reviewer 2 Report

Popova et al reported their work named "The relevance of implementing the systematic screening of perioperative myocardial injury in noncardiac surgery patients " and concluded "The implementation of screening for PMI is highly recommended, at least in high-risk patients. However, there are challenges and complexities involved in achieving its optimal implementation". I have the following comments:

- Please specify the primary outcome (eg MACE) in your methods and try to add a multivariate predictor of this outcome during the first month as the current Table 1 represents only univariate analysis. 

- Table 3: Please report the number then the percentage between brackets and not the reverse.

- Please try to add figures to your manuscript eg barplot of the percentage of different outcomes during 1st month vs after.

- Please consider adding more relevant papers t your discussion eg https://www.sciencedirect.com/science/article/pii/S2772930323000182

Minor language revision is needed

Author Response

Dear editors,

Thank you for the opportunity to resubmit a revised version of our manuscript. We have considered the comments of the reviewers and the editorial team and provided a point-to-point reply below with our comments in italics followed by questions in normal font. We also took opportunity and have corrected some orthographic and grammar spelling throughout the manuscript, as well as updated references order. The changes in the manuscript are highlighted as required.

Reviewer 2

Popova et al reported their work named "The relevance of implementing the systematic screening of perioperative myocardial injury in noncardiac surgery patients " and concluded "The implementation of screening for PMI is highly recommended, at least in high-risk patients. However, there are challenges and complexities involved in achieving its optimal implementation".

We sincerely thank the reviewer for the interest and constructive suggestions and recommendations. Following these advices, we have now made according changes at various parts of the main text, tables, and figures.

I have the following comments:

- Please specify the primary outcome (eg MACE) in your methods and try to add a multivariate predictor of this outcome during the first month as the current Table 1 represents only univariate analysis. 

Regarding the primary outcome MACCE (page 5, lines 201 and 206), also mentioned by the Reviewer 1, we updated its definition as follows: "Our primary outcome was to assess the incidence of death and major cardiovascular and cerebrovascular events (MACCE) within one month and one year after surgery."

Regarding the second part of the question about multivariate predictor which also was mentioned by the Reviewer 1, we requested advice from our statistical advisor. Following his recommendation, we decided not to implement this analysis as doing so would lead to the creation of numerous models, and consequently, collinearity problems might arise frequently.

- Table 3: Please report the number then the percentage between brackets and not the reverse.

We accordingly updated the table 3 putting the number then the percentage between brackets. However, as we added the bar-plot (Figure 3), following your suggestion in the next observation, in order to not extend the graphical content, the Table 3 is now presented as supplementary table 4. We included information about these changes at the main text at page 10, line 358.

- Please try to add figures to your manuscript eg barplot of the percentage of different outcomes during 1st month vs after.

Thank you for the suggestion. We have added a bar-plot (as Figure 3) comparing MACCE frequency between the two-time sets analyzed in the study.

- Please consider adding more relevant papers t your discussion eg https://www.sciencedirect.com/science/article/pii/S2772930323000182

Thank you for the interesting suggestion, we accordingly included information about this relevant paper at discussion section (lines: 504-507). Additionally, we added it at references as number 37.

Comments on the Quality of English Language

Minor language revision is needed.

Submission Date

16 July 2023

Date of this review

01 Aug 2023 13:17:53

Round 2

Reviewer 1 Report

Thank you for responding to the issues raised during the peer-review process.